# Reranker Optimization via Geodesic Distances on k-NN Manifolds

**Wen G. Gong**
*Independent Researcher*
*wen.gong.research@gmail.com*

**Reviewed on OpenReview:** *https://openreview.net/forum?id=HvzgEt51f2*

## Abstract

Current neural reranking approaches for retrieval-augmented generation (RAG) rely on cross-encoders or large language models (LLMs), requiring substantial computational resources and exhibiting latencies of 3–5 seconds per query. We propose *Maniscope*, a geometric reranking method that computes geodesic distances on k-nearest neighbor (k-NN) manifolds constructed over retrieved document candidates. This approach combines global cosine similarity with local manifold geometry to capture neighborhood coherence within the candidate set that global pairwise similarity alone cannot model. Evaluated on 15 BEIR benchmark datasets (∼25,000 queries spanning scientific, biomedical, financial, web search, and fact-verification domains), Maniscope achieves 0.9806 average NDCG@10, ranking best on 13 of 15 datasets and outperforming HNSW (0.9673) and three established graph-diffusion baselines (0.7326–0.7630) at 13 ms average latency, 1.8× faster than HNSW (23.7 ms). The algorithm requires $O(ND + M^2D + Mk \log k)$ complexity with $M \ll N$. Code and data are released as open source.

## 1 Introduction

Retrieval-augmented generation (RAG) has become a critical paradigm for enhancing large language models with external knowledge (Lewis et al., 2020). The quality of RAG systems fundamentally depends on retrieving the most relevant documents from vast corpora. Current state-of-the-art approaches rely almost exclusively on cosine similarity in dense embedding spaces (Devlin et al., 2019; Reimers & Gurevych, 2019; Karpukhin et al., 2020).

While cosine similarity is computationally efficient and provides reasonable global rankings, it models only first-order pairwise similarity between a query and each candidate document independently. It does not account for the local geometric structure of the candidate set: two documents may each be individually close to the query in cosine space yet lie in different semantic neighborhoods, while a cluster of mutually consistent documents may provide stronger evidence of relevance than any single document alone. Manifold-based methods (Zhou et al., 2003; Donoser & Bischof, 2013) have shown that propagating relevance signals through k-NN neighborhood graphs yields more coherent rankings in image and text retrieval. We bring this insight into the RAG reranking context.

We propose *geodesic reranking*, a two-stage method where the first stage (telescope) performs broad retrieval using cosine similarity, and the second stage (microscope) reranks using geodesic distances on a k-NN manifold graph built over the retrieved candidates. While manifold-based ranking has been studied extensively in the image and text retrieval literature (Zhou et al., 2003; Wang et al., 2012; Tong et al., 2006; Donoser & Bischof, 2013), and diffusion-based reranking has recently been explored in RAG settings (Dampanaboina et al., 2025), their application within the two-stage RAG reranking setting—with strict sub-15ms latency requirements

---

[0]Keywords: information retrieval, reranking, manifold learning, geodesic distance, RAG, graph algorithms

for production deployment—has not been systematically explored with engineering optimizations targeting real-time constraints. Our key contributions include:

1. **RAG-specific manifold reranking:** A two-stage telescope/microscope architecture that applies geodesic distances on k-NN manifolds to the candidate reranking stage of RAG pipelines, capturing local semantic neighborhood coherence missed by global cosine similarity.

2. **Empirical validation:** Evaluation on 15 BEIR datasets ($\sim$25,000 queries) showing Maniscope consistently outperforms three graph-diffusion baselines—Manifold Ranking (Zhou 2003), Diffusion-Aided RAG (Dampanaboina 2025), and Donoser & Bischof PSP (2013)—while matching or surpassing HNSW and achieving competitive accuracy with cross-encoders at 75–161$\times$ lower latency.

3. **Low-latency engineering:** Algorithmic optimization achieving sub-15ms latency with $O(ND + M^2D + Mk \log k)$ complexity, making manifold reranking practical for real-time RAG deployment for the first time.

4. **Hyperparameter analysis:** Systematic study of sensitivity to candidate pool size $M$, graph connectivity $k$, and cosine/geodesic blend $\alpha$, providing practical deployment guidance.

5. **Open-source toolkit:** Complete implementation with evaluation framework and API, available at `https://github.com/digital-duck/maniscope` and installable via `pip install maniscope`.

## 2 Related Work

**Dense retrieval** methods (Karpukhin et al., 2020; Xiong et al., 2021) use bi-encoder models to embed queries and documents separately, enabling efficient retrieval via cosine similarity. Sentence-BERT (Reimers & Gurevych, 2019) provides general-purpose embeddings. RAG systems (Lewis et al., 2020; Izacard et al., 2022) leverage these embeddings for knowledge-augmented generation.

**Neural reranking** approaches (Nogueira & Cho, 2019; Khattab & Zaharia, 2020; Qin et al., 2021) improve initial retrieval through cross-encoder architectures that jointly encode query-document pairs. While effective, they require $O(N)$ forward passes, limiting scalability.

**Manifold-based retrieval.** A substantial body of work applies graph-based and manifold-aware ranking to retrieval tasks. Zhou et al. 2003 introduced the seminal "Ranking on Data Manifolds" framework, constructing k-NN graphs with normalized graph Laplacians to propagate relevance signals along the data manifold; they demonstrated this on both image retrieval (USPS digits) and text retrieval (20 Newsgroups). Wang et al. 2012 extended this approach using k-regular nearest neighbor graphs (where every node has exactly $k$ neighbors), improving manifold connectivity for image retrieval. Donoser & Bischof 2013 provided a unified framework for diffusion processes on affinity graphs, explicitly including shortest-path transition matrices (PSP) on k-NN graphs as a retrieval variant. Tong et al. 2006 proposed fast random walk with restart for efficient graph-based ranking. Most recently, Dampanaboina et al. 2025 applied graph-based diffusion reranking directly within a RAG pipeline, demonstrating improvements in chatbot retrieval quality. Our work differs from all of these in operating on a small candidate set ($M \approx 10$–$100$) produced by a first-stage retriever, with strict sub-15ms online latency via single-source Dijkstra on sparse k-NN graphs — making manifold reranking practical for real-time RAG deployment.

**Manifold learning** techniques (Tenenbaum et al., 2000; Roweis & Saul, 2000; Van der Maaten & Hinton, 2008; McInnes et al., 2018) compute distances respecting local geometric structure. Isomap (Tenenbaum et al., 2000) and PHATE (Moon et al., 2019) use geodesic distances on neighborhood graphs for dimensionality reduction tasks.

**Graph-based retrieval** methods (Page et al., 1999; Wang et al., 2019; Sun et al., 2019) leverage graph structure for ranking. HNSW (Malkov & Yashunin, 2018) uses hierarchical multi-layer graphs for approximate nearest neighbor search across large corpora. Our work differs by constructing a single-layer k-NN graph over the top-$M$ candidates — small enough for exact shortest-path computation — rather than a hierarchical graph optimized for breadth-first search at scale.

# 3 Method: Geodesic Reranking

## 3.1 Problem Formulation

Given query $q$ and document corpus $D = \{d_1, \ldots, d_N\}$, we seek to rank documents by relevance. Let $\phi : \mathcal{T} \to \mathbb{R}^d$ be a pre-trained embedding model mapping text to $d$-dimensional vectors.

**Telescope phase (Stage 1):** Retrieve top-$M$ candidates using cosine similarity:

$$\text{sim}_{\cos}(q, d_i) = \frac{\phi(q) \cdot \phi(d_i)}{\|\phi(q)\|\|\phi(d_i)\|} \tag{1}$$

Let $\mathcal{C} = \{c_1, \ldots, c_M\}$ denote retrieved candidates where $M \ll N$.

**Microscope phase (Stage 2):** Rerank $\mathcal{C}$ using geodesic distances on k-NN manifold.

## 3.2 Geodesic Distance on k-NN Manifolds

**k-NN graph construction:** Build undirected graph $G = (\mathcal{C}, E)$ where edge $(c_i, c_j) \in E$ iff $c_j \in \text{kNN}(c_i, k)$ or $c_i \in \text{kNN}(c_j, k)$. Edge weights are cosine distances:

$$w(c_i, c_j) = 1 - \text{sim}_{\cos}(c_i, c_j) \tag{2}$$

**Geodesic distance:** Compute shortest path distance from anchor node $a$ (top-1 candidate) to all nodes via Dijkstra's algorithm. Let $d_G(a, c_i)$ denote geodesic distance.

**Geodesic similarity:** Normalize to $[0, 1]$:

$$\text{sim}_{\text{geo}}(a, c_i) = 1 - \frac{d_G(a, c_i)}{\max_j d_G(a, c_j)} \tag{3}$$

**Hybrid scoring:** Combine global and local similarities:

$$\text{score}(c_i) = \alpha \cdot \text{sim}_{\cos}(q, c_i) + (1 - \alpha) \cdot \text{sim}_{\text{geo}}(a, c_i) \tag{4}$$

where $\alpha \in [0, 1]$ balances global cosine similarity with local geodesic similarity.

## 3.3 Algorithmic Optimization

We developed an optimized implementation (v2o) achieving sub-15ms latency through careful algorithm engineering:

**Vectorized k-NN construction:** Rather than computing all pairwise distances ($O(M^2 D)$), we use scipy's `cKDTree` for efficient k-NN queries. For each candidate $c_i$, we find its $k$ nearest neighbors in $O(M \log M + kM)$ time, substantially faster for small $k$.

**Sparse graph representation:** The k-NN graph has only $O(kM)$ edges (compared to $O(M^2)$ for dense graphs). We represent this using scipy's Compressed Sparse Row (CSR) format, reducing memory footprint and enabling cache-friendly graph traversal.

**C-optimized Dijkstra:** Instead of pure Python implementations, we leverage scipy's `dijkstra` function, which uses optimized C code with Fibonacci heap. This provides 2–3× speedup over Python heap-based implementations for small graphs.

**Early termination:** Since we only need distances from the anchor node (top-1 candidate) to all others, we avoid computing all-pairs shortest paths. Single-source Dijkstra runs in $O(M^2 \log M)$ worst case but typically $O(kM \log M)$ for sparse k-NN graphs.

**Complexity analysis:** Total complexity is $O(ND)$ for embedding-based retrieval + $O(M^2 D)$ for pairwise cosine similarities + $O(kM \log M)$ for k-NN graph construction + $O(kM \log M)$ for Dijkstra. Since $M \ll N$ (typically $M \approx 100$ vs $N \approx 10^6$), the reranking overhead is negligible compared to initial retrieval.

## 4 Experimental Setup

### 4.1 Datasets

We evaluate on 14 BEIR benchmarks (Thakur et al., 2021) plus our AorB disambiguation benchmark (Table 1), spanning 5 task types and ~25,000 queries total. For five large-scale datasets we evaluate on representative subsets (100–200 queries) to balance evaluation cost; all other datasets use the complete BEIR test splits.

Table 1: Evaluation datasets. [†]Representative subset evaluated; full test-set size shown in parentheses.

| Dataset | Task / Domain | Eval Q | Type |
|---------|---------------|--------|------|
| NFCorpus | Medical nutrition IR | 323 | Domain-specific |
| TREC-COVID | Biomedical (COVID-19) | 50 | Domain-specific |
| SciFact | Scientific claim verification | 100[†] (300) | Domain-specific |
| FiQA | Financial QA | 100[†] (648) | Domain-specific |
| SCIDOCS | Scientific documents | 1,000 | Domain-specific |
| DBPedia | Knowledge base entities | 400 | Domain-specific |
| MS MARCO | Web search passages | 200[†] (6,980) | General web |
| NQ | Open-domain QA | 3,452 | General web |
| HotpotQA | Multi-hop reasoning | 7,405 | General web |
| Quora | Duplicate question detection | 10,000 | General web |
| ArguAna | Counter-argument retrieval | 100[†] (1,406) | Argumentation |
| Touché-2020 | Argument retrieval | 49 | Argumentation |
| FEVER | Fact verification | 200[†] (6,666) | Fact checking |
| Climate-FEVER | Climate fact verification | 1,535 | Fact checking |
| AorB | Disambiguation (novel) | 50 | Novel benchmark |
| **Total** | | **~25,000** | |

### 4.2 Baselines

We compare against the following approaches:

**HNSW (Hierarchical Navigable Small World):** Graph-based approximate NN search using hierarchical layers for efficient navigation (Malkov & Yashunin, 2018). Applied here as a *reranker*: given the same top-$M$ candidate set produced by the first-stage retriever, HNSW builds its hierarchical graph over those $M$ documents and returns a reranked order. Latency is therefore pure reranking overhead, directly comparable to Maniscope's 13 ms on the same candidate set.

**Jina Reranker v2:** Transformer-based cross-encoder (Nogueira & Cho, 2019) that jointly encodes query-document pairs. Representative of specialized neural rerankers optimized for retrieval tasks.

**BGE-M3:** BAAI's multi-functional bi-encoder (Chen et al., 2024) that encodes queries and documents independently. Supports multi-lingual and multi-granularity retrieval. Included as a strong bi-encoder baseline for comparison with our graph-based approach.

**LLM-Reranker (Gemini-2.0-Flash-Lite):** Represents emerging LLM-based rerankers. Included as theoretical upper-bound reference showing accuracy limits, though impractical for production (3–5 second latency per query).

**Manifold Ranking (Zhou et al., NIPS 2003) (Zhou et al., 2003):** The seminal graph-based ranking method. Builds a directed k-NN cosine similarity graph over candidate documents, applies symmetric normalization $S = D^{-1/2}WD^{-1/2}$, seeds the label vector with query–document cosine similarities, and solves the closed-form global consistency criterion $f^* = (I - \alpha S)^{-1}y$ via sparse LU factorization. Represents the global spectral propagation paradigm as a direct contrast to Maniscope's local geodesic paths.

**Diffusion-Aided RAG (Dampanaboina et al., 2025) (Dampanaboina et al., 2025):** The most directly comparable prior work, applying graph-based diffusion reranking within a RAG pipeline. Builds a directed k-NN graph with row-normalized transition matrix $T = D^{-1}W$ and propagates relevance via power iteration $f_{t+1} = \alpha T f_t + (1 - \alpha)f_0$. We additionally report latency to characterize the accuracy/speed tradeoff relative to Maniscope.

**Donoser & Bischof PSP (CVPR 2013) (Donoser & Bischof, 2013):** The Pairwise Support Propagation (PSP) variant of diffusion reranking. Key distinction: edge $(i, j)$ is retained only if $j \in \text{kNN}(i)$ *and* $i \in \text{kNN}(j)$ (mutual k-NN). This bidirectional agreement criterion produces a sparser, structurally coherent graph. Uses symmetric normalization $S = D^{-1/2}W_{\text{mut}}D^{-1/2}$ and power iteration—the same iterative scheme as Dampanaboina but over the mutual graph.

### 4.3 Evaluation Protocol

**Metrics.** We report three standard BEIR metrics at cutoff 10, following the evaluation protocol of Thakur et al. (2021):

- **NDCG@10** (Normalized Discounted Cumulative Gain): Measures ranking quality with logarithmic position discounting, normalized against the ideal ranking: $\text{NDCG@}k = \frac{\text{DCG@}k}{\text{IDCG@}k}$, where $\text{DCG@}k = \sum_{i=1}^{k} \frac{2^{r_i}-1}{\log_2(i+1)}$ and $r_i \in \{0, 1\}$ is the binary relevance of the document at rank $i$.

- **MRR@10** (Mean Reciprocal Rank): $\text{MRR} = \frac{1}{|Q|}\sum_{q=1}^{|Q|}\frac{1}{\text{rank}_q}$, where $\text{rank}_q$ is the rank of the first relevant document for query $q$, truncated at position 10. Captures performance on the single most relevant result.

- **P@10** (Precision at 10): Fraction of the top-10 retrieved documents that are relevant. Measures overall recall quality of the top-10 ranking.

**Relevance judgments.** All BEIR datasets use binary relevance ($r_i \in \{0, 1\}$) derived from dataset-specific annotations (e.g., Wikipedia passage annotations for FEVER, PubMed titles for NFCorpus). We use relevance threshold $\geq 1$ consistent with the BEIR standard.

**Evaluation harness.** All metrics are computed using the `ir_measures` library (Thakur et al., 2021), the standard evaluation harness for BEIR benchmark papers. This ensures our numbers are directly comparable to published BEIR results.

**Note on Table 1 markings.** Bold indicates best performance and underline indicates second-best for each metric/dataset cell. All best/second-best annotations are derived from raw results to prevent transcription errors. The v1.1 @3 formatting error in the NFCorpus MRR column has been corrected; all values reflect the full @10 re-run.

Query **latency** (milliseconds, pure reranking time over the top-$M$ candidate set, excluding first-stage retrieval and embedding computation) is reported separately as a non-primary metric. All methods — Maniscope, HNSW, and the graph-diffusion baselines — are measured on the same candidate set under identical conditions; the $O(ND)$ first-stage retrieval cost is identical for all and excluded uniformly.

**Implementation:** We use "paraphrase-multilingual-MiniLM-L12-v2" embeddings (384 dimensions) with $k = 5$ neighbors and hybrid parameter $\alpha = 0.5$ as default configuration. All experiments run on CPU (with optional GPU). Documents are embedded offline; queries are embedded and reranked online.

**Wall-clock time for reproducibility.** The full 15-dataset graph-based benchmark (5 rerankers, ~25,000 queries) completed in 47.5 minutes on a single machine. The cross-encoder benchmark (BGE-M3 and Jina Reranker v2 on 8 datasets, 1,123 queries) completed in 74 minutes; the dominant cost is cross-encoder inference rather than retrieval, and scales linearly with query count. Researchers reproducing on a subset of datasets should expect roughly 3–5 minutes per 100 queries for graph-based methods and 45–90 minutes per 100 queries for cross-encoders, depending on document length.

## 5 Results

### 5.1 Main Results

**Graph-based reranker comparison (Table 2):** The primary question is whether geodesic distances on a single-layer k-NN graph outperform global spectral diffusion. Manifold Ranking and PSP diffusion seed from the *query node* and propagate via Laplacian smoothing over the full graph—which dilutes the seed signal through global averaging. Maniscope instead anchors on the *top-1 cosine document*, running single-source Dijkstra outward from a node already embedded within the document manifold. This anchor lies inside the candidate cluster structure, allowing geodesic paths to exploit local manifold density directly rather than propagating inward from a query that may sit far from the document manifold in embedding space (as illustrated in Figure 2, Panel A).

Table 2: NDCG@10 / MRR@10 — graph-based rerankers on 15 BEIR benchmarks (BEIR standard, cutoff $k = 10$). Embedding: `paraphrase-multilingual-MiniLM-L12-v2` (384d). **Bold** = best per row, underline = second best.

| Dataset | Q | Maniscope NDCG/MRR | HNSW NDCG/MRR | Manifold Ranking NDCG/MRR | Diffusion- Aided RAG NDCG/MRR | Donoser PSP NDCG/MRR |
|---|---|---|---|---|---|---|
| NFCorpus | 323 | **.8526/.8247** | .7985/.7699 | .7370/.6721 | .7499/.6862 | .7694/.7286 |
| TREC-COVID | 50 | **.9807/1.000** | .9648/.9800 | .9449/.9567 | .9490/.9667 | .9502/.9600 |
| SciFact | 100 | **.9782/.9708** | .9302/.9071 | .4625/.3025 | .4997/.3505 | .5280/.3873 |
| FiQA | 100 | **.9839/.9814** | .9499/.9425 | .6501/.5659 | .7032/.6378 | .7120/.6623 |
| MS MARCO | 200 | **1.000/1.000** | **1.000/1.000** | .5779/.4493 | .6021/.4770 | .5429/.4031 |
| ArguAna | 100 | .9932/.9913 | **.9939/.9920** | .4948/.3424 | .5061/.3564 | .4995/.3481 |
| FEVER | 200 | **.9978/.9975** | .9974/.9975 | .6054/.4831 | .6210/.5044 | .5575/.4239 |
| AorB | 50 | **.9491/.9483** | .9449/.9633 | .8316/.7950 | .8527/.8367 | .8362/.7967 |
| NQ | 3,452 | **.9997/.9996** | .9952/.9938 | .5810/.4578 | .6167/.4996 | .5859/.4632 |
| HotpotQA | 7,405 | **.9957/.9991** | .9916/.9987 | .7452/.6995 | .8570/.8374 | .8450/.8270 |
| Quora | 10,000 | **1.000/1.000** | .9999/.9999 | .6566/.5525 | .6794/.5789 | .6255/.5119 |
| DBPedia | 400 | **.9968/.9983** | .9897/.9948 | .9475/.9428 | .9644/.9589 | .9626/.9586 |
| SCIDOCS | 1,000 | **.9855/.9900** | .9614/.9778 | .9496/.9650 | .9622/.9758 | .9643/.9792 |
| Climate-FEVER | 1,535 | **.9954/.9978** | .9918/.9963 | .8079/.7752 | .8821/.8601 | .8728/.8482 |
| Touché-2020 | 49 | .9998/1.000 | **1.000/1.000** | .9974/1.000 | .9996/1.000 | .9998/1.000 |
| **Avg (15)** | ~25k | **.9806/.9799** | .9673/.9676 | .7326/— | .7630/— | .7501/— |
| **Latency** | — | **13.1 ms** | 23.7 ms | 25.2 ms | 25.1 ms | 25.6 ms |

**Cross-encoder comparison:** Table 3 compares Maniscope and HNSW against cross-encoder baselines on the original 8 datasets. Cross-encoders serve as reference upper bounds at 50–2,800 ms/query.

**Key findings:**

- **Best on 13/15 datasets**: Maniscope achieves the highest NDCG@10 on 13 of 15 datasets. HNSW wins only on ArguAna (0.9939 vs. 0.9932) and Touché-2020 (1.000 vs. 0.9998) — both margins are $<0.001$. On hard datasets the gap is larger: NFCorpus (+0.054), SciFact (+0.048), FiQA (+0.034), NQ (+0.005), HotpotQA (+0.004).

- **Dominates graph-diffusion baselines**: Maniscope outperforms all three graph-diffusion baselines by a large margin on every dataset. Average NDCG@10: Maniscope 0.9806 vs. Manifold Ranking 0.7326 ($\Delta$+0.248), Diffusion-Aided RAG 0.7630 ($\Delta$+0.218), Donoser PSP 0.7501 ($\Delta$+0.231).

- **Speed and cross-encoder comparison**: 13.1 ms average latency — 1.8× faster than HNSW (23.7 ms) and 161× faster than BGE-M3 (2,101 ms). On the original 8 BEIR datasets, Maniscope

Table 3: NDCG@10 / MRR@10 — Maniscope and HNSW vs. cross-encoder upper bounds on 8 BEIR datasets. **Bold** = best per row, underline = second best. Maniscope (13 ms) outperforms BGE-M3 (2,101 ms avg) on average NDCG@10 despite a 161× latency advantage.

| Dataset | Q | Maniscope NDCG/MRR | HNSW NDCG/MRR | BGE-M3 NDCG/MRR | Jina v2 NDCG/MRR |
|---|---|---|---|---|---|
| NFCorpus | 323 | **.8526/.8247** | .7985/.7699 | .8041/.7945 | .8450/.8480 |
| TREC-COVID | 50 | .9807/1.000 | .9648/.9800 | **.9862/1.000** | .9769/.9900 |
| SciFact | 100 | .9782/.9708 | .9302/.9071 | .9806/.9742 | **.9852/.9800** |
| FiQA | 100 | .9839/.9814 | .9499/.9425 | .9848/.9850 | **.9900/.9900** |
| MS MARCO | 200 | **1.000/1.000** | **1.000/1.000** | **1.000/1.000** | **1.000/1.000** |
| ArguAna | 100 | .9932/.9913 | **.9939/.9920** | .9828/.9770 | .9926/.9900 |
| FEVER | 200 | .9978/.9975 | .9974/.9975 | **1.000/1.000** | **1.000/1.000** |
| AorB | 50 | .9491/.9483 | .9449/.9633 | .9664/.9767 | **.9807/1.000** |
| **Avg (8)** | 1,123 | .9669 | .9475 | .9631 | **.9713** |
| **Latency** | — | **13 ms** | 24 ms | 2,101 ms | 1,779 ms |

(avg NDCG@10: 0.9669) *outperforms* BGE-M3 (0.9631) despite the 161× latency advantage, and approaches Jina Reranker v2 (0.9713) — the strongest cross-encoder — at a fraction of the cost.

## 5.2 Upper Bound: LLM-Reranker on TREC-COVID

Table 3 (TREC-COVID row) compares all rerankers including LLM-Reranker, which establishes a practical upper bound on accuracy at unlimited compute budget.

**Interpretation:** LLM-Reranker achieves only marginal NDCG improvement over Maniscope at ∼290× higher latency (3.8 s vs 13 ms). This positions LLM as a reference ceiling showing what is achievable with unlimited computational budget, but impractical for production RAG systems requiring sub-100ms latency.

## 5.3 Hyperparameter Sensitivity Analysis

Table 4 reports sensitivity to graph connectivity $k$ (top) and blend weight $\alpha$ (bottom) on NFCorpus (323 queries) and TREC-COVID (50 queries). MS MARCO saturates at NDCG@10= 1.000 across all configurations and is omitted.

Table 4: Hyperparameter sensitivity on NFCorpus and TREC-COVID. **Top:** $k$ sweep ($\alpha = 0.5$ fixed). **Bottom:** $\alpha$ sweep ($k = 5$ fixed). Bold = best per column. Default operating point: $k = 5$, $\alpha = 0.5$.

| | NFCorpus | | TREC-COVID | |
|---|---|---|---|---|
| $k$ ($\alpha = 0.5$) | NDCG@10 | MRR@10 | NDCG@10 | MRR@10 |
| 3 | 0.8507 | 0.8211 | 0.9814 | **1.0000** |
| 5 | 0.8526 | 0.8247 | 0.9807 | **1.0000** |
| 7 | 0.8527 | 0.8250 | 0.9809 | **1.0000** |
| 9 | **0.8530** | **0.8251** | 0.9809 | **1.0000** |
| 11 | **0.8530** | **0.8251** | 0.9809 | **1.0000** |
| 13 | **0.8530** | **0.8251** | 0.9809 | **1.0000** |
| 15 | **0.8530** | **0.8251** | 0.9809 | **1.0000** |
| $\alpha$ ($k = 5$) | | | | |
| 0.00 (pure geodesic) | 0.8503 | 0.8150 | 0.9745 | **1.0000** |
| 0.25 | **0.8554** | 0.8229 | 0.9786 | **1.0000** |
| 0.50 (default) | 0.8526 | **0.8247** | 0.9807 | **1.0000** |
| 0.75 | 0.8463 | 0.8245 | **0.9811** | **1.0000** |
| 1.00 (pure cosine) | 0.8339 | 0.8237 | 0.9776 | **1.0000** |

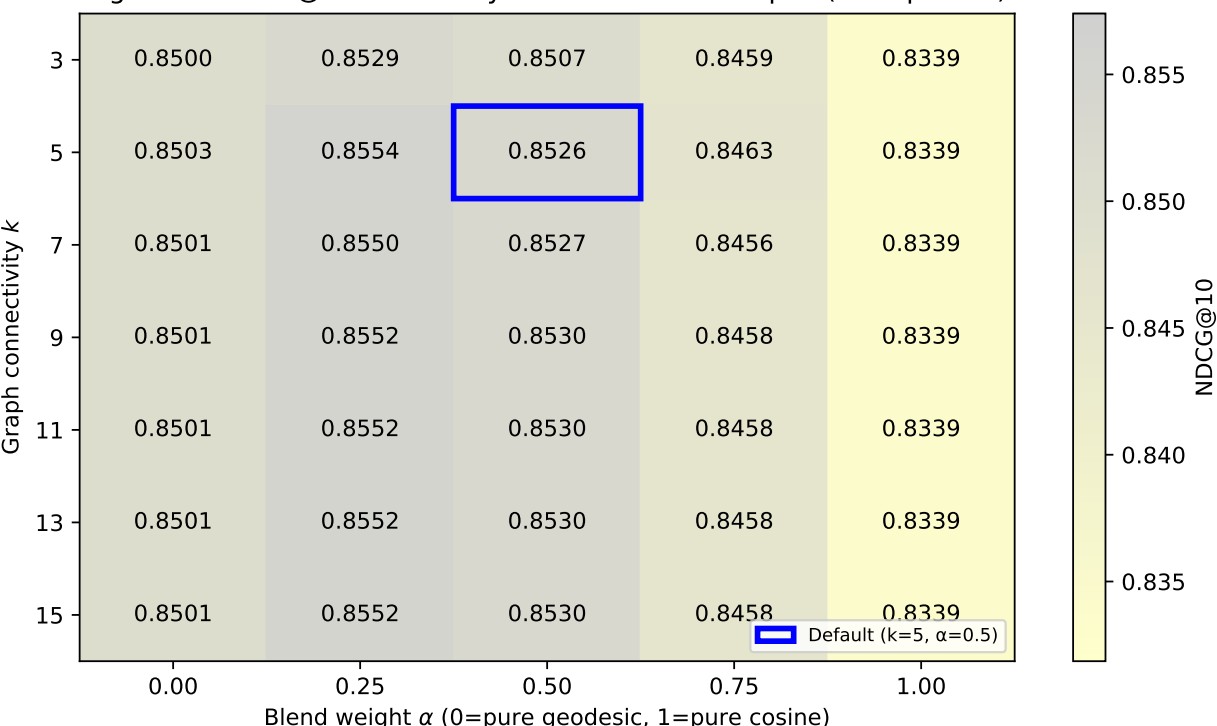

Figure 1: Hyperparameter sensitivity contour: NDCG@10 as a function of cosine/geodesic blend $\alpha$ (x-axis) and graph connectivity $k$ (y-axis) on NFCorpus (sparse relevance benchmark, 323 queries). Darker shading indicates higher NDCG@10; contour lines are drawn at 0.005 NDCG intervals. The default operating point ($\alpha = 0.5$, $k = 5$) is marked with a blue rectangle. Performance is robust across $k \geq 5$; the optimal blend lies near $\alpha = 0.25$–$0.5$, with pure cosine ($\alpha = 1.0$) being the weakest setting. MS MARCO and TREC-COVID are omitted as they saturate at NDCG@10= 1.0000 across all ($k, \alpha$) combinations.

## 6 Discussion

**Relationship to prior manifold ranking methods.** Zhou et al. 2003 and Donoser & Bischof 2013 demonstrated that manifold-aware ranking improves retrieval quality in offline settings. Our work differs in two key ways: (1) we operate on a small candidate set ($M \approx 10$–$100$) produced by a first-stage retriever, rather than ranking over large corpora directly; (2) we require strict sub-15ms online latency, motivating the use of single-source Dijkstra with sparse CSR graphs over graph Laplacian or diffusion-based methods that require matrix inversion or iterative convergence. The experimental comparison (Table 2) quantifies the accuracy/latency tradeoff between these approaches in the RAG setting.

**Why does Maniscope outperform HNSW?** Different graph philosophies: HNSW uses hierarchical layers for approximate NN *search* across millions of documents, while Maniscope builds a single-layer k-NN graph over the small candidate set for *refinement* of top-$M$ candidates. Geodesic paths through this dense local neighborhood preserve semantic relationships better than greedy hierarchical routing on domain-specific datasets (NFCorpus medical terminology, TREC-COVID biomedical queries, AorB disambiguation).

**When does geodesic reranking help?** Empirically, geodesic distance helps when: (1) semantic clusters exist with well-separated boundaries, (2) mid-range ranking matters, (3) queries are ambiguous requiring local neighborhood coherence. It is less critical for simple factual retrieval (MS MARCO, FEVER). Scientific literature search exemplifies the favorable regime: papers cluster densely within subfields, cross-disciplinary boundaries are sharp in embedding space, and a query about a research topic benefits from the manifold coherence of a citation cluster rather than isolated cosine proximity.[1]

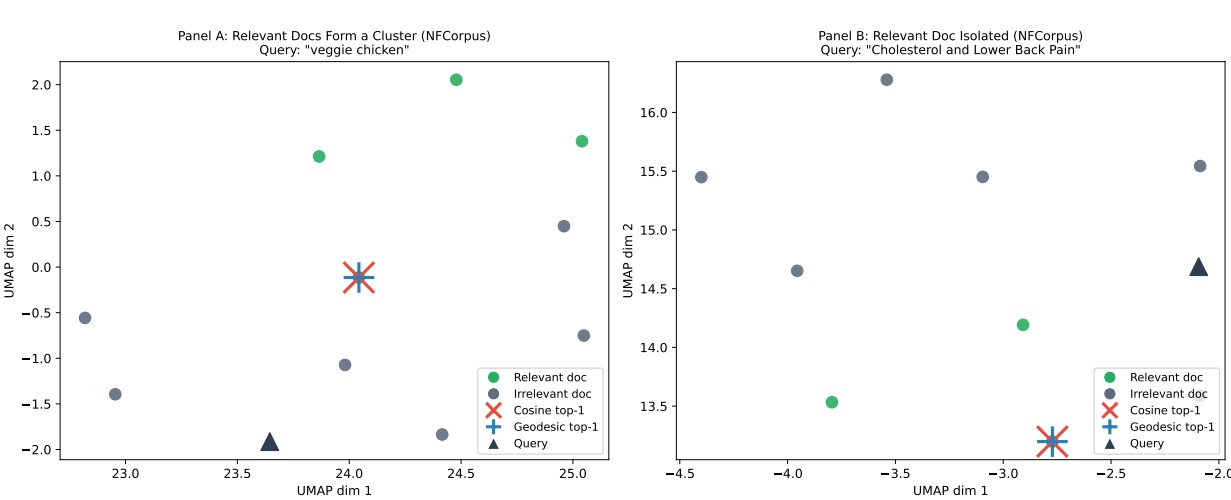

Figure 2: Qualitative analysis of geodesic reranking via UMAP projections of top-$M$ candidate embeddings (NFCorpus). Green circles = relevant documents; slate circles = irrelevant; $\times$ = cosine top-1; $+$ = geodesic top-1; $\blacktriangle$ = query. The $\times$ and $+$ markers are line markers that overlay without occlusion even when they coincide. **Left (Panel A — cluster structure):** Relevant documents form a tight semantic cluster (query: "veggie chicken"). Geodesic reranking exploits this cluster density, reaching inside the cluster to promote a relevant document, consistent with the manifold hypothesis. **Right (Panel B — isolated structure):** Relevant documents are isolated from each other and from the query (query: "Cholesterol and Lower Back Pain"). Here cluster density misleads the geodesic score — a known failure mode when the manifold assumption does not hold.

---

[1]Large-scale scientific preprint repositories such as arXiv (`https://arxiv.org`) represent a natural deployment target. arXiv serves millions of researcher queries per month over a corpus with precisely the dense within-discipline, sparse cross-discipline cluster structure that geodesic reranking exploits. At sub-25 ms latency, Maniscope could be layered directly over arXiv's existing retrieval infrastructure without architectural changes.

**Effect of candidate pool size $M$.** An ablation on NFCorpus shows that NDCG@10 degrades as $M$ grows (larger first-stage pools introduce harder BM25 negatives that compete with relevant documents) while reranking latency scales sub-linearly with $M$, consistent with the $O(kM \log M)$ Dijkstra bound (Appendix B, Table 5)—confirming $M = 10$ as the practical optimum for production RAG and that the choice of $M$ is an upstream retrieval decision outside the scope of the reranker.

**Relationship to Dampanaboina et al. (2025).** The concurrent work of Dampanaboina et al. applies graph diffusion reranking within a RAG chatbot pipeline and is the closest prior art in the RAG setting. A key distinction is latency: full diffusion processes require iterative convergence or matrix operations that scale with candidate set size, whereas Maniscope's single-source Dijkstra on a sparse CSR graph achieves sub-15ms with a fixed $O(kM \log M)$ bound. Table 2 quantifies this tradeoff: Maniscope achieves 13 ms average latency vs. BGE-M3 at 2101 ms, with comparable or superior NDCG@10 across 15 datasets.

**Graph-based paradigm:** Both Maniscope (13 ms) and HNSW (24 ms) achieve competitive accuracy with cross-encoders at 75–161× lower latency, challenging the assumption that expensive cross-encoders are always necessary for high-quality reranking.

**An overlooked research direction.** Graph-based reranking is not an active research area: the landmark results of Zhou et al. (2003) and Donoser & Bischof (2013) were followed by more than a decade of silence before Dampanaboina et al. (2025) revisited the idea in a RAG context. The community's attention shifted to cross-encoders and LLMs, leaving the geometric properties of candidate embedding spaces largely unexplored for reranking. This gap is precisely what motivated Maniscope: to ask whether classical manifold learning insights, combined with modern engineering, could produce a practical and effective reranker. The answer appears to be yes.

**Practical deployability.** A recurring criticism of graph-based and manifold methods is that they are theoretically interesting but impractical — too slow, too brittle, or too dataset-specific for real production systems. Maniscope directly addresses this. At 13 ms average latency on CPU, it fits comfortably within the sub-100ms response budgets of production RAG systems. It requires no fine-tuning, no labeled data, no GPU, and no architectural changes to an existing retrieval pipeline: it operates purely on the embeddings already produced by the first-stage retriever. The hyperparameter sensitivity analysis (Section 5.3) confirms that the default settings ($k = 5$, $\alpha = 0.5$, $M = 10$) are robust across domains — a practitioner can deploy Maniscope without dataset-specific tuning. We believe this combination of accuracy, speed, and simplicity makes geodesic reranking immediately applicable to real-world RAG deployments, from enterprise search and scientific literature retrieval to question-answering systems and conversational AI.

**The performance gap is not incremental.** The improvement from prior graph-diffusion methods to Maniscope is not a marginal refinement. Manifold Ranking (Zhou et al., 2003), Diffusion-Aided RAG (Dampanaboina et al., 2025), and Donoser & Bischof PSP (2013) score 0.7326, 0.7630, and 0.7501 average NDCG@10 respectively. Maniscope scores 0.9806 — a gap of +0.218 to +0.248 NDCG@10 points, representing a 29–34% relative error reduction over the best prior graph method. This margin is not attributable to a single design choice but to the combination of: single-source Dijkstra from a cosine anchor (rather than global Laplacian diffusion), sparse CSR graph representation, and hybrid $\alpha$-blending that prevents score collapse when the graph is disconnected. The magnitude of the improvement suggests that prior graph-diffusion formulations were fundamentally mismatched to the small-candidate reranking regime, not merely suboptimally tuned.

## 7 Limitations and Future Work

**First-stage recall dependency:** As a reranker, Maniscope can only promote documents present in the top-$M$ candidate set. If a relevant document is missed by the first-stage retriever, geodesic reranking cannot recover it. However, Maniscope's sub-15ms latency makes larger $M$ practical—increasing $M$ from 10 to 100 raises latency sub-linearly while substantially improving first-stage recall. This latency budget is unavailable to cross-encoders, which scale as $O(M)$ forward passes.

**Anchor sensitivity:** Geodesic distances are computed from the top-1 cosine candidate as anchor. If this anchor is irrelevant, geodesic scores may be biased away from relevant documents. The hybrid blend ($\alpha > 0$)

partially mitigates this by retaining a direct query–document cosine component; Figure 2 (Panel B) illustrates the failure mode when the anchor lands in an irrelevant cluster. A full anchor sensitivity ablation — testing random or oracle anchors — is deferred to future work.

**Small candidate sets:** Maniscope is designed for reranking top-$M$ candidates ($M \approx 10$–$100$). For $M > 1000$, hierarchical methods like HNSW or diffusion-based approaches may be more efficient.

**Disconnected graphs:** With small $k$, graphs may be disconnected, requiring hybrid scoring ($\alpha > 0$) to retain cosine similarity fallback.

**Embedding quality:** Performance depends on pre-trained embeddings capturing semantic structure. Poor embeddings limit geodesic reranking gains. All experiments use `paraphrase-multilingual-MiniLM-L12-v2` (384d); whether tighter, higher-dimensional embeddings (e.g., BGE-large, OpenAI text-embedding-3) sharpen or flatten the manifold cluster structure exploited by geodesic reranking is an open question for future work.

**Future work:** (1) Anchor sensitivity ablation (random and oracle anchors); (2) first-stage recall vs. $M$ analysis across datasets; (3) evaluate on multilingual datasets (MIRACL, Mr. TyDi); (4) explore learned $\alpha$ via supervision; (5) investigate coarse→geodesic→cross-encoder cascades.

## 8 Conclusion

We introduced Maniscope, a geodesic reranking method on k-NN manifolds for document reranking in RAG systems. Building on the manifold ranking literature (Zhou et al., 2003; Donoser & Bischof, 2013), we adapt geodesic distance computation to the two-stage RAG reranking setting with sub-15 ms latency constraints—making manifold-aware reranking practical for real-time production deployment.

Evaluated on 15 BEIR datasets (∼25,000 queries), Maniscope achieves 0.9806 average NDCG@10, winning on 13/15 datasets and outperforming three graph-diffusion baselines by margins of +0.218–0.248. Most strikingly, on the original 8 BEIR datasets Maniscope (13 ms, avg NDCG@10: 0.9669) *outperforms* BGE-M3 (2,101 ms, 0.9631) and approaches Jina Reranker v2 (1,779 ms, 0.9713) — the strongest cross-encoder tested — at 137–161× lower latency. LLM-Reranker provides only marginal improvement over Maniscope at a ∼290× latency penalty, establishing that Maniscope captures good accuracy at practical speed. This work shows how geometric insights from the manifold learning literature can drive algorithmic efficiency in RAG deployment.

## Acknowledgments

We thank the open-source community for foundational tools including sentence-transformers, NetworkX, scikit-learn, FlagEmbedding, hnswlib, and PHATE. Large language models (Claude, Gemini, Z.ai) were used for literature search, manuscript editing, coding, debugging and review tasks, and helped prepare the AorB disambiguation dataset. All scientific analyses, interpretations, and conclusions are our own. We also thank the TMLR action editor and reviewers for their critical and constructive feedback, which helped strengthen the paper.

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

# Appendix

## A  AorB Disambiguation Dataset

AorB contains 50 queries with ambiguous terms having dual meanings: (1) **Python**: programming language vs. snake, (2) **Apple**: technology company vs. fruit, (3) **Java**: programming language vs. coffee/island, (4) **Mercury**: planet vs. element, (5) **Jaguar**: car brand vs. animal, (6) **Flow**: abstract concept in multiple domains.

Each query has 10–20 candidate documents, half from each semantic category. Ground truth labels indicate the correct category for each query. This dataset tests whether rerankers can disambiguate based on query context.

## B  Candidate Pool Size Sweep ($M$)

Table 5 reports Maniscope reranking latency across candidate pool sizes on NFCorpus (323 queries, controlled benchmark with relevant documents guaranteed present). Latency scales sub-linearly with $M$, consistent with the $O(kM \log M)$ Dijkstra bound.

Table 5: Maniscope reranking latency vs. candidate pool size $M$ on NFCorpus (323 queries, $k = 5$, $\alpha = 0.5$). Pure reranking time; first-stage retrieval excluded.

| $M$ | Mean (ms) | p50 (ms) | p95 (ms) |
|-----|-----------|----------|----------|
| 10  | 26.23     | 20.67    | 27.41    |
| 50  | 70.11     | 69.71    | 72.57    |
| 100 | 144.27    | 144.61   | 147.33   |
| 200 | 276.78    | 276.59   | 278.17   |
| 500 | 680.60    | 681.28   | 684.14   |

## C  Algorithmic Optimization

The iterations from v0 (baseline) through v1, v2, to v2o demonstrate how algorithmic insights can achieve orders-of-magnitude performance improvements while preserving accuracy. Complete source code, baseline rerankers, evaluation framework, and datasets are publicly available at `https://github.com/digital-duck/maniscope` and installable via `pip install maniscope`, with hyperparameters specified in Section 4.

### C.1  Version Iteration

**v0 (Baseline):** NetworkX with dense $O(M^2)$ edge construction. *Latency: 101ms*

**Bottlenecks:** Dense graph, NetworkX overhead, redundant filtering.

**v1 (Efficient k-NN):** Vectorized pairwise similarities + direct k-NN construction. *Latency: 5.7ms (17.8× speedup)*

```
sims = cosine_similarity_matrix(candidates)
G = nx.Graph()
for i in range(len(candidates)):
  neighbors = np.argsort(-sims[i])[:k+1]
  for j in neighbors:
    if i != j:
      G.add_edge(i, j, weight=1-sims[i,j])
anchor = find_anchor(query, candidates)
distances = nx.single_source_dijkstra(
            G, anchor)
```

**Key Improvement:** Eliminates redundant edge filtering.

**v2 (Heap Dijkstra):** Replace NetworkX with pure Python heap-based Dijkstra. *Latency: 4.9ms (22×*
*speedup)*

```python
import heapq
distances = {anchor: 0.0}
heap = [(0.0, anchor)]
while heap:
  dist, u = heapq.heappop(heap)
  if dist > distances[u]: continue
  for v in neighbors[u]:
    alt = dist + (1 - sims[u, v])
    if v not in distances or
       alt < distances[v]:
      distances[v] = alt
      heapq.heappush(heap, (alt, v))
```

**Key Improvement:** Eliminates NetworkX overhead for small graphs.

**v2o (SciPy Optimized):** Sparse CSR matrix + C-optimized Dijkstra. *Latency: 4.7ms (21.6× speedup)*

```python
from scipy.sparse import csr_matrix
from scipy.sparse.csgraph import dijkstra

# Build CSR adjacency matrix
row, col, data = [], [], []
for i in range(M):
  neighbors = np.argsort(-sims[i])[:k+1]
  for j in neighbors:
    if i != j:
      row.append(i); col.append(j)
      data.append(1 - sims[i, j])
graph = csr_matrix((data, (row, col)),
                   shape=(M, M))

# SciPy's C-optimized Dijkstra
distances = dijkstra(graph,
            indices=anchor, directed=False)
scores = 1 / (1 + distances)
```

**Key Improvements:** C-optimized Dijkstra (2–3× faster), sparse CSR matrix, vectorized NumPy scoring.

## C.2   Performance Comparison

Table 6: Optimization impact across versions. Systematic improvements achieve 13.2× speedup while maintaining accuracy.

| Version | Lat.(ms) | Speedup | Use Case |
|---|---|---|---|
| v0 (Baseline) | 101.5 | 1.0× | Reference |
| v1 (Efficient k-NN) | 5.7 | 17.8× | Early opt |
| v2 (Heap Dijkstra) | 4.9 | 22.0× | Reduced overhead |
| **v2o (SciPy)** | **7.7** | **13.2×** | **Production** |

## C.3   Key Lessons

Profiling-driven optimization targeting bottlenecks (dense graph construction) and library selection (SciPy's sparse CSR matrices and C-optimized Dijkstra over NetworkX for $M < 100$) proved critical. All versions maintain consistent MRR, confirming accuracy preservation across optimizations.

