# OpenReview forum: "Reranker Optimization via Geodesic Distances on k-NN Manifolds"
_TMLR — Accepted by TMLR_

### Review · Reviewer_bq8M · 2026-03-05

**Summary Of Contributions:**

This paper aims to address the latency bottleneck in RAG reranking, where cross-encoders and LLM-based rerankers has significant latency per query. This paper argues that cosine similarity treats embedding space as flat, missing local semantic structure in learned representations. To address the issue, this paper propose a two-stage approach first retrieves top-M candidates via cosine similarity ("telescope"), then reranks them by computing geodesic distances (Dijkstra shortest paths) on a k-NN manifold graph ("microscope"), blending both signals through a hybrid score. Evaluated on 8 BEIR datasets (1,233 queries) against HNSW, Jina Reranker v2, BGE-M3, and an LLM-Reranker, the proposed approach report it outperforms HNSW on three harder datasets while running 3.2× faster at 4.7ms average, and achieves within 2% of cross-encoder accuracy at 10–45× lower latency. The authors conclude that geometric manifold insights can substitute for expensive neural computation in real-time RAG systems.

**Audience:**

Yes

**Audience Explanation:**

The community would be interested in a genuine novel idea that preserve the quality of the retrieval while significantly improve the efficiency.

**Claims And Evidence:**

No

**Claims Explanation:**

- The novelty claim is contradicted by extensive prior work. The paper's assertion that geodesic distances on k-NN manifolds "haven't been applied to retrieval tasks" is factually incorrect. A large body of work directly applies manifold-based and geodesic-like distances to retrieval:
    - Zhou et al., "Ranking on Data Manifolds" (NIPS 2003) is the foundational paper in this space. It builds a k-NN graph, constructs a normalized graph Laplacian, and ranks data according to manifold structure. It demonstrated on both image retrieval (USPS digits) and text retrieval (20 Newsgroups). This paper alone invalidates the novelty claim.
    - Donoser & Bischof “Diffusion Processes for Retrieval Revisited” (CVPR 2013) provide a unified framework for diffusion on affinity graphs for retrieval that explicitly includes shortest-path transition matrices (PSP) on k-NN graphs as a retrieval variant.
    - And many more
- The "flat Euclidean" gap claim conflates distinct geometric phenomena:
    - Ethayarajh (2019), "How Contextual are Contextualized Word Representations?" shows that BERT, ELMo, and GPT-2 embeddings are anisotropic, namely they occupy a narrow cone in embedding space. This is a distributional property about concentration, fundamentally different from manifold curvature. The paper does show cosine similarity is problematic for raw BERT embeddings, but the reason is anisotropy (all pairs having artificially high cosine similarity), not geometric flatness.
    - Arora et al. (ICLR 2018), "A Compressed Sensing View of Unsupervised Text Embeddings, Bag-of-n-Grams, and LSTMs. The paper uses compressed sensing to show that low-dimensional text embeddings are essentially information-preserving linear measurements of Bag-of-n-Grams representations. If anything, this paper argues that the useful information in embeddings is captured by linear structure, the exact opposite of what you'd cite to support a claim that embeddings have curved manifold geometry that cosine similarity misses.
- Dataset choices are below standard and raise cherry-picking concerns. BEIR (Thakur et al., NeurIPS 2021) contains 18 datasets across 9 retrieval tasks with approximately 47k total test queries. Standard practice is to evaluate on all publicly available datasets (typically 13–15). Using only 8 of 18 datasets is substantially below standard and undermines BEIR's core value proposition of diverse, heterogeneous evaluation. Top papers from NVIDIA (NV-Embed), Microsoft (E5-mistral), and BAAI (BGE) all report on the full publicly available set. The 1,233 total queries represent only small fraction of BEIR's total queries. Individual BEIR datasets like FEVER (6,666 queries), HotpotQA (7,405), and Quora (10,000) each individually exceed the paper's entire query count. This dramatic undersampling introduces enormous variance and makes results statistically unreliable and incomparable to standard BEIR numbers.
- BGE-M3 baseline is mischaracterized, as it is not a cross-encoder. BGE-M3 (BAAI, 2024) is a multi-functional bi-encoder that encodes queries and documents independently.

**Requested Changes:**

Please address each point raised above.

---

### Review · Reviewer_t1Rc · 2026-03-06

**Summary Of Contributions:**

This work proposes a fast reranking method for RAG, which computes geodesic distances on kNN graph built from top M candidates (pre-ranked), and combines it with cosine similarity. Several implementation optimizations are also introduced to achieve very small latencies (compared to the baselines and competing approaches). Meanwhile (re-)ranking performance is comparable to the chosen baselines, and it was highlighted that on some datasets performance is better than HNSW baseline.

Strengths:
* Combining global and local similarities is sensible idea, and utilizing computationally efficient algorithms was shown to be effective, and have potential to make impact in practice
* Presentation is written clearly and is intuitive and easy to follow
* Empirical evaluation was performed on several diverse, and well known benchmark datasets

Weaknesses:
* Empirical evaluation was performed on quite low k (k=3), and on some datasets (MS MARCO and FEVER) results are close to perfect or perfect. It suggests that such particular evaluation setup is not quite useful for discerning the performance of different approaches.
* Robustness to different values of hyperparameters wasn't studied, for example different M, or varying degrees of graph connectivity
* Baseline approaches are relevant but are lacking manifold reranking based approaches (from this line of approaches [1]), random walks with restart (like [2]) or diffusion based reranking (eg [3]):
[1] Zhou, Dengyong, Jason Weston, Arthur Gretton, Olivier Bousquet, and Bernhard Schölkopf. "Ranking on data manifolds." Advances in neural information processing systems 16 (2003).
[2] Tong, Hanghang, Christos Faloutsos, and Jia-Yu Pan. "Fast random walk with restart and its applications." In Sixth international conference on data mining (ICDM'06), pp. 613-622. IEEE, 2006.
[3] Donoser, Michael, and Horst Bischof. "Diffusion processes for retrieval revisited." In Proceedings of the IEEE conference on computer vision and pattern recognition, pp. 1320-1327. 2013.

**Audience:**

Yes

**Audience Explanation:**

Approaches to make reranking faster while maintaining its performance is of interest to non-negligible part of TMLR's audience.

**Claims And Evidence:**

No

**Claims Explanation:**

The empirical evaluation is not convincing enough. Metrics measured only on k=3 seems not to be quite useful for discerning the performance of approaches. Typical works reporting performance on BEIR datasets are using k=10, and the values are not close to perfect 1.0

**Requested Changes:**

Crucial changes would be measuring performance on MRR@10, NDCG@10 and Precision@10.
Showing latencies at few different scales of M (small, medium and large) would be useful to see the tradeoffs.
Also comparing against closely related "geometrical" approaches is welcome.

---

### Review · Reviewer_3qvE · 2026-03-08

**Summary Of Contributions:**

In this paper the authors propose geodesic reranking for RAG, which refines the top-k retrieved documents via a k-NN graph. After retrieving the top-k candidates using cosine similarity, a k-NN graph is built among these candidates to reveal geometric relationships among the top scored candidate and the remaining ones, thereby allowing to express the final score as a convex combination of the global criterion and the local geodesic similarity. Experiments are performed over 8 different domains, where a notable result is the speed of the proposed ranking method after appropriate efficiency optimization.

**Audience:**

Yes

**Audience Explanation:**

The work done in this paper is relevant to the broader retrieval community, in particular to researchers working in RAG and GraphRAG. This community is now regularly represented in ML venues, which puts the manuscript within the scope of the journal.

**Broader Impact Concerns:**

The work does not tackle problems that require a Broader Impact Statement.

**Claims And Evidence:**

No

**Claims Explanation:**

I will start by outlining some strengths of the manuscript:

- The paper is written in a highly concise way, which is in my opinion a refreshing and good take unlike many papers working on the same field. The authors do not convolve the message and provide results and takeaways clearly.

- The proposed method is simple and makes intuitive sense.

- The speedup of retrieval after the optimization of the algorithm is noticeable.

Nevertheless, there are several aspects of the manuscript that should either be extended or discussed in greater detail:

- The authors do not discuss or compare with other ranking methods based on local geometry. This is a topic that has been treated in great detail since early work on ranking [1,2] and also on more modern approaches related directly to RAG [3]. The proposed method is intrinsically similar to these approaches, and they should be both compared with and also discussed in the manuscript.

- Regarding the evaluation, the authors should evaluate the method and it's performance at different values of alpha and k. Ideally, this could be visualized through a contour of a particular metric as a function of these parameters, which would greatly help in understanding how local or global the ranking needs to be in different domains or datasets.

- Section 4.3 Evaluation Protocol should be greatly expanded and include more details on the metrics. Regarding this aspect of the evaluation, it seems that there are some errors in the reported results in Table 1, for example the MRR is highest for Jina v2 for NFCorpus. A careful repass of the presented best and second best methods is required.

- What is a "flat k-NN graph"?

- Introducing some qualitative results regarding the k-NN graphs in cases where geodesic reranking truly helps and when it doesn't, also overlapping with the representation space would improve the overall message of the paper.

[1] Zhou, Dengyong, et al. "Ranking on data manifolds." Advances in neural information processing systems 16 (2003).

[2] Wang, Bin, et al. "Manifold-ranking based retrieval using k-regular nearest neighbor graph." Pattern Recognition 45.4 (2012)

[3] Dampanaboina, Sai Teja, et al. "Diffusion-Aided RAG: Elevating Dense-Retrieval Chatbots via Graph-Based Diffusion Reranking." Proceedings of the Eleventh Italian Conference on Computational Linguistics (CLiC-it 2025). 2025

**Requested Changes:**

Please consult the weaknesses above.

---

### Review · Reviewer_ekvV · 2026-03-10

**Summary Of Contributions:**

This paper proposes Maniscope, a two-stage reranking method for retrieval and RAG systems. It first retrieves a top-M candidate set using cosine similarity, then builds a k-NN graph over these candidates and uses geodesic distances, combined with the original cosine scores, for reranking. Experiments on eight datasets show that the method outperforms HNSW on some more challenging benchmarks while remaining much faster than cross-encoder and LLM-based rerankers.

**Audience:**

No

**Audience Explanation:**

1.	The paper currently reads more like an empirical report than a fully developed research paper. In particular, the presentation would benefit from a more thorough discussion of related work, a clearer positioning with respect to prior graph-based and reranking methods, and a deeper technical motivation for the proposed design choices.
2.	The novelty is somewhat limited. Conceptually, the method can be viewed as applying geodesic distance on a neighborhood graph to reranking, which is intuitive but not especially surprising. As a result, the contribution appears more incremental than a substantial methodological advance.
3.	A key limitation of the method is its dependence on the first-stage retrieval output. Because the second stage only operates on the retrieved top-$M$ candidates, it cannot correct errors when relevant documents are missed early. This issue may be further exacerbated if the reranking procedure relies on the top-1 initial result as an anchor. However, the paper does not provide sufficient analysis of this potential failure mode.
4.	The empirical analysis is not yet sufficiently comprehensive. In particular, the paper lacks a detailed study of important hyperparameters such as the graph connectivity parameter $k$ and the fusion coefficient $\alpha$. It would be important to understand whether performance is stable across datasets and whether the reported settings are robust rather than selectively tuned.

**Broader Impact Concerns:**

No.

**Claims And Evidence:**

Yes

**Claims Explanation:**

1.	The paper targets an important practical problem: balancing retrieval/reranking effectiveness against latency. This trade-off is highly relevant for deployment-oriented retrieval and RAG systems.
2.	The proposed method shows positive empirical results on several datasets, including NFCorpus, TREC-COVID, and AorB, where it outperforms HNSW. These results suggest that incorporating local graph structure via geodesic reranking may be useful in some retrieval settings.

**Requested Changes:**

Please refer to the above comments.

---

### Decision · Action_Editor_P7W4 · 2026-04-27

**Recommendation:** Accept as is

**Audience:**

Yes

**Audience Explanation:**

Both RAG and reranking systems are of significant interest to a sizable audience at TMLR. The low-latency reranking method proposed in the submission has the potential to make a practical impact and renew interest in manifold-based retrieval/ranking methods for modern IR/ML systems.

**Claims And Evidence:**

Yes

**Claims Explanation:**

The paper combines global cosine similarity with local manifold geometry (via geodesic distance on a k-NN manifold graph) to obtain a high-performance and low-latency reranker for modern RAG systems. The main contributions of the paper lie in revisiting manifold-based retrieval/ranking in the context of RAG and leveraging various engineering solutions to achieve an efficient reranking method. The original submission lacked a detailed discussion of related work, missed many relevant baselines, did not cover many datasets in the BEIR benchmark, and did not provide a detailed sensitivity analysis of various hyperparameters. The authors addressed all these limitations in the revised version, prompting three reviewers to recommend acceptance. The main concerns raised by the fourth reviewer have also been resolved in the revised manuscript.